# Hephaestus Minicubes:
# A Global, Multi-Modal Dataset for Volcanic Unrest Monitoring

**Nikolas Papadopoulos** [1]
npapadopoulos@mail.ntua.gr

**Nikolaos Ioannis Bountos**[1,2]
bountos@noa.gr

**Maria Sdraka**[1,2]
masdra@noa.gr

**Andreas Karavias**[1]
andreas_karavias@mail.ntua.gr

**Ioannis Papoutsis**[1]
ipapoutsis@mail.ntua.gr

[1] **Orion Lab**
National Observatory of Athens & National Technical University of Athens

[2] **Harokopio University of Athens**

## Abstract

Ground deformation is regarded in volcanology as a key precursor signal preceding volcanic eruptions. Satellite-based Interferometric Synthetic Aperture Radar (InSAR) enables consistent, global-scale deformation tracking; however, deep learning methods remain largely unexplored in this domain, mainly due to the lack of a curated machine learning dataset. In this work, we build on the existing *Hephaestus* dataset, and introduce *Hephaestus Minicubes*, a global collection of 38 spatiotemporal datacubes offering high resolution, multi-source and multi-temporal information, covering 44 of the world's most active volcanoes over a 7-year period. Each spatiotemporal datacube integrates InSAR products, topographic data, as well as atmospheric variables which are known to introduce signal delays that can mimic ground deformation in InSAR imagery. Furthermore, we provide expert annotations detailing the type, intensity and spatial extent of deformation events, along with rich text descriptions of the observed scenes. Finally, we present a comprehensive benchmark, demonstrating *Hephaestus Minicubes'* ability to support volcanic unrest monitoring as a multi-modal, multi-temporal classification and semantic segmentation task, establishing strong baselines with state-of-the-art architectures. This work aims to advance machine learning research in volcanic monitoring, contributing to the growing integration of data-driven methods within Earth science applications.

## 1 Introduction

Ground deformation monitoring plays a vital role in volcanic hazard assessment, providing early insights into subsurface magmatic activity [Dzurisin, 2003]. It is widely regarded as one of the most reliable eruption precursors, with detectable signals that may emerge from days to even years before an event [Biggs et al., 2014]. Timely detection of such signals can offer critical lead time for risk mitigation and emergency response efforts [Tilling, 2008].

While ground-based networks, particularly those relying on Global Navigation Satellite Systems (GNSS), have traditionally been used to monitor deformation [Poland, 2024], many volcanoes

Submitted to 39th Conference on Neural Information Processing Systems (NeurIPS 2025). Do not distribute.

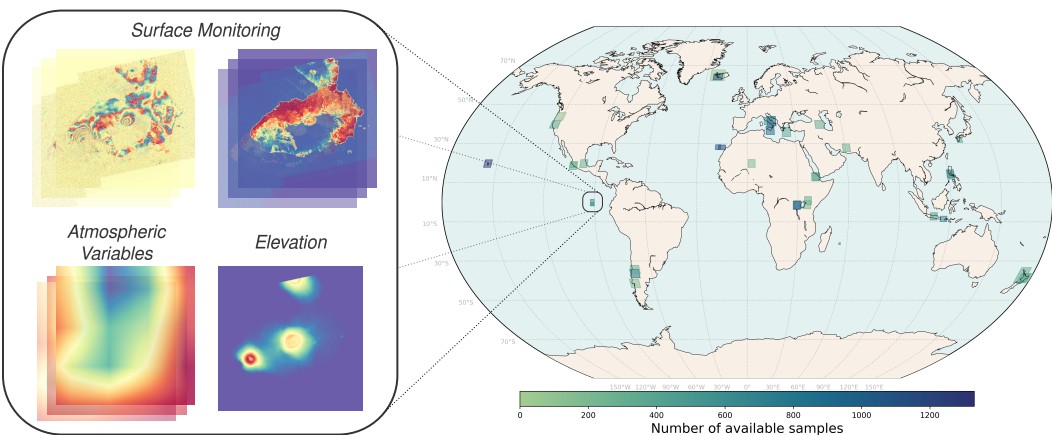

Figure 1: Hephaestus Minicubes data sources (left) vis-à-vis the spatial distribution of the minicubes (right). Box sizes on the map are proportional to frame dimensions and color intensity reflects the number of available products per region.

worldwide remain poorly instrumented or entirely unmonitored Loughlin et al. [2015]. This limitation, coupled with the growing availability of publicly accessible satellite data, from missions such as Copernicus Sentinel, has prompted a shift toward satellite-based approaches Spaans and Hooper [2016]. Among these, Interferometric Synthetic Aperture Radar (InSAR) has emerged as a powerful tool for global monitoring of surface motion [Hanssen, 2001].

InSAR estimates surface displacement with millimeter-level precision by analyzing phase differences between two or more SAR acquisitions from the same location at different times, while coherence quantifies the similarity between signals, serving as a measure of phase reliability and surface stability. A major challenge in interpreting InSAR data is distinguishing true ground deformation from atmospheric propagation delays. Lateral variations in ionospheric electron density and tropospheric water vapor concentration can alter the radar signal's propagation time, introducing phase delays that contaminate the InSAR deformation signal [Zebker et al., 1997, Massonnet and Feigl, 1998, Beauducel et al., 2000]. These delays can produce artifacts that mimic real deformation, sometimes manifesting as apparent centimeter-scale ground motion [Doin et al., 2009], thereby complicating data interpretation and downstream analysis. The issue is even more prominent in volcanic regions, where complex atmospheric conditions—especially vertical stratification in mountainous terrain—can generate deformation-like patterns. This increases the risk of false positives in unrest detection, especially over elevated topography such as volcanoes and high-altitude ridges [Parker et al., 2015, Shirzaei and Bürgmann, 2012].

Deep learning pipelines have been successfully developed for various SAR-based tasks in Earth observation [Zhu et al., 2021], including natural disasters mapping (*e.g.* flood mapping Bountos et al. [2025]). However, their application to InSAR remains limited, mainly due to the lack of a curated machine learning dataset. This poses a significant barrier as the processing, understanding and annotation of InSAR products require specialized domain expertise. *Hephaestus* [Bountos et al., 2022c] marked the first attempt to construct a unique, expert-annotated InSAR dataset centered around volcanic activity monitoring. Despite its contributions, *Hephaestus* exhibits several limitations, which we discuss in detail in section 3.1, that have hindered its broader adoption within the community.

Our work builds on the *Hephaestus* dataset, by enhancing it both in ground sampling distance (GSD), as well as in information depth, by introducing additional data sources, and engineering its structure to better support time-series analysis. *Hephaestus Minicubes* introduces a collection of high-resolution spatiotemporal datacubes integrating InSAR phase difference and coherence products, digital elevation models (DEMs), and relevant atmospheric variables known to confound deformation signals (Fig. 1). In addition, we include a diverse set of expert annotations characterizing deformation type, intensity, and extent. Leveraging these improvements, we establish a comprehensive benchmark demonstrating the ability of *Hephaestus Minicubes* to support volcanic unrest monitoring as a multi-modal, multi-temporal classification and semantic segmentation task. We report strong baselines

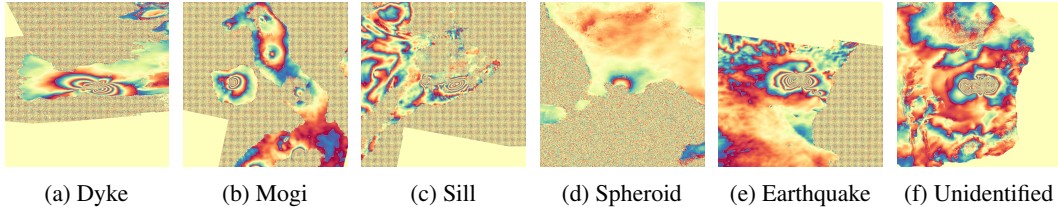

| (a) Dyke | (b) Mogi | (c) Sill | (d) Spheroid | (e) Earthquake | (f) Unidentified |

Figure 2: Examples of the different ground deformation types available in *Hephaestus Minicubes*.

using state-of-the-art architectures, while also identifying key limitations and challenges associated with applying deep learning in this context.

To support further research and promote the application of machine learning in InSAR-based volcanic unrest monitoring, we publicly release the *Hephaestus Minicubes* dataset at https://github.com/Orion-AI-Lab/Hephaestus-minicubes. The repository includes comprehensive documentation and is actively maintained to provide access to the latest version of the dataset. All code is released under the MIT License [1] and data under the CC-BY license [2].

## 2   Related Work

Despite the recent success of deep learning in Earth Observation (*e.g.* [Sumbul et al., 2021, Sdraka et al., 2024]), its adoption in the InSAR domain has remained limited. One of the main reasons for this is the lack of a large curated dataset, primarily due to a) the scarcity of positive instances and b) the high cost of the annotation process, which demands expert knowledge.

To overcome these challenges, and alleviate the need for time-consuming manual annotation, many works leveraged synthetically generated datasets, and models pretrained on optical tasks. In particular, [Anantrasirichai et al., 2018], relied on major data augmentations and transfer learning from ImageNet [Deng et al., 2009]. Building on this, several works focused on synthetically generated InSAR data to train Convolutional Neural Networks (CNNs) for ground deformation detection *e.g.* [Brengman and Barnhart, 2021, Anantrasirichai et al., 2019] and [Gaddes et al., 2024]. [Valade et al., 2019] utilized synthetic data to train a CNN to predict the associated phase gradients and phase decorrelation mask, which can later be used to detect ground displacement. Similarly, [Beker et al., 2023] utilized a synthetic dataset to train CNNs to detect subtle ground deformation from velocity maps. [Bountos et al., 2022a] proposed to train transformer architectures on synthetically generated InSAR using a prototype learning framework, assigning classes with a nearest-neighbor approach comparing the sample's representation with the class prototypes.

Bountos et al. [2022b] diverged from this line of research and proposed the utilization of in-domain self-supervised contrastive learning to create reliable feature extractors without the need for human annotations, emphasizing the performance improvement compared to pretrained weights from optical tasks. In a separate line of work, [Popescu et al., 2024] proposed to formulate the volcanic ground deformation identification problem as an anomaly detection task utilizing Patch Distribution Modeling [Defard et al., 2021].

Given the gradual evolution of volcanic activity, time-series analysis is critical for effective monitoring, with several works exploring this direction. [Sun et al., 2020] trained a CNN on synthetic data, generated using the Mogi model as the deformation source. They created 20,000 time-series groups, with each group containing 20 time-consecutive pairs of unwrapped surface displacement maps. Moreover, [Ansari et al., 2021] proposed an unsupervised pipeline to cluster similar displacement patterns from InSAR time-series, building on a Long Short Term Memory (LSTM) autoencoder and Hierarchical Density-Based Spatial Clustering of Applications with Noise (HDBSCAN) Campello et al. [2013].

Finally, [Bountos et al., 2022c] introduced *Hephaestus* in an attempt to address the data scarcity at its core. *Hephaestus* was the largest manually annotated InSAR dataset to date with global coverage. Its introduction addressed many open gaps enabling the deployment of large deep learning models on a

---

[1]https://opensource.org/license/MIT
[2]https://creativecommons.org/licenses/by/4.0/

variety of ground deformation related problems, while paving the way for the adaptation of complex multi-modal tasks to the InSAR domain *e.g.* InSAR captioning and text to InSAR generation. Despite its significance, however, *Hephaestus* still presents notable limitations, which we discuss in detail in Section 3.1.

## 3 Dataset Construction

### 3.1 Building on Hephaestus

The *Hephaestus* dataset represents a significant step towards advancing machine learning-based approaches for volcanic unrest monitoring. While it offers rich, expert annotations across a global set of volcanoes, its effectiveness in high-precision and time-series geophysical analysis is limited by several factors. First, the spatial resolution of the annotated imagery is relatively coarse, at approximately $333\,\text{m} \times 333\,\text{m}$ per pixel. Second, the dataset consists of RGB composites of the InSAR products, lacking physically interpretable pixel values and geolocation information. Finally, the dataset structure is not designed for spatiotemporal modeling, lacking a machine-learning-friendly format. Recognizing both the promise and the limitations of *Hephaestus*, we take steps to redefine the dataset by addressing its weaknesses and expanding its scope.

### 3.2 Hephaestus Minicubes

In this work we introduce *Hephaestus Minicubes*, a collection of 38 datacubes covering 44 of the most active volcanoes globally from 2014 to 2021, with a significantly enhanced spatial resolution of approximately $100\,\text{m} \times 100\,\text{m}$ per pixel, containing a total of 19,942 annotated samples. Each datacube integrates InSAR products, topographic information, atmospheric variables that are known to introduce delays to SAR signals, combined under a diverse collection of expert annotations. The datacubes are stored in a compressed `Zarr` format [Miles et al., 2020], as structured multi-dimensional arrays optimized for efficient spatiotemporal analysis, with the full dataset totaling 1.7 TB.

In the following paragraphs, we provide a detailed description of each component of the *Hephaestus Minicubes* dataset, along with important design choices made during its development.

**InSAR Products.** The InSAR component lies at the core of the dataset, including: a) the wrapped *phase difference*, which captures surface displacement between SAR acquisitions, and b) the *coherence*, which measures the quality of the interferometric signal. These products are acquired by the COMET-LiCSAR system, which processes Copernicus Sentinel-1 imagery for global volcano surveillance, in a resolution of approximately $100\,\text{m} \times 100\,\text{m}$ per pixel. For more information on the InSAR generation and processing pipeline, readers are referred to [Lazecký et al., 2020].

**Topography.** Stratified atmospheric noise is often correlated with topography. To capture this we include the *Digital Elevation Model (DEM)* from LiCSAR, based on the 1 arc-second Shuttle Radar Topography Mission *DEM* [Farr et al., 2007]. This static variable is downsampled for each frame to match the resolution of the InSAR products, approximately $100\,\text{m} \times 100\,\text{m}$.

**Atmospheric Component.** A key advancement of *Hephaestus Minicubes* is the explicit integration of atmospheric variables known to directly contribute to phase delays in the SAR signal. These delays may produce patterns in InSAR imagery that closely resemble true surface deformation, [Zebker et al., 1997, Massonnet and Feigl, 1998, Beauducel et al., 2000] Following state-of-the-art atmospheric correction methods [Yu et al., 2018a] we incorporate atmospheric variables that represent humidity, temperature, and pressure. Specifically we include *Total Column Water Vapour*, *Surface Pressure*, and the *Vertical Integral of Temperature*, from the ERA5 single-level reanalysis dataset [Hersbach et al., 2020], for both primary and secondary SAR acquisition dates. We prioritize vertically integrated atmospheric measures, as the impact of atmospheric delays is not confined to specific atmospheric

Table 1: Summary of annotated activity variables

| Annotation Variable | Count |
| --- | --- |
| **Label** | |
| Non-Deformation | 18089 |
| Deformation | 1798 |
| Earthquake | 55 |
| **Activity Type** | |
| Sill | 1258 |
| Dyke | 527 |
| Mogi | 333 |
| Earthquake | 55 |
| Unidentified | 50 |
| Spheroid | 25 |
| **Intensity Level** | |
| Low | 908 |
| Medium | 533 |
| High | 751 |
| None | 55 |
| **Phase** | |
| Rest | 18089 |
| Unrest | 1664 |
| Rebound | 134 |
| None | 15 |

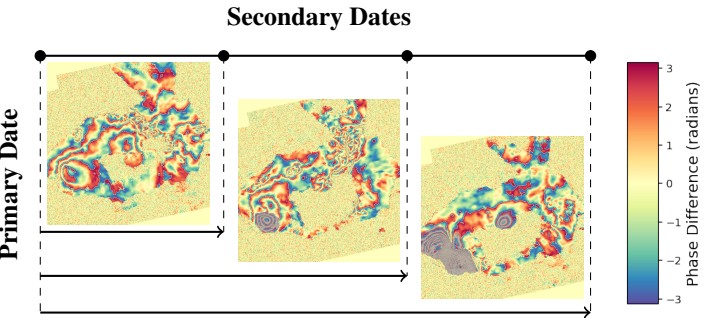

Figure 3: Schematic representation of the time-series construction method. A single primary acquisition date is associated with multiple secondary dates, forming a sequence of InSAR products. Each image displays the phase difference with an overlaid mask to highlight areas with apparent deformation.

layers. We select the ERA5 data closest in time to each SAR acquisition and resample them to align with the spatial resolution of the InSAR data.

**Expert Annotations.** *Hephaestus Minicubes* builds upon the manually curated annotations provided by *Hephaestus*, adapting them to the datacube format by converting relevant labels into spatiotemporal masks and carefully addressing differences in spatial resolution and alignment. The available spatiotemporal masks include multi-level information on *ground deformation*, *activity type* (*e.g.* Dyke, Sill, Spheroid, etc.), and *intensity level* (*e.g.* Low, Medium, High), while the related volcano's *phase* (*e.g.* Rest, Unrest, Rebound) is represented as a categorical variable (see Tab.1). Additional annotations provide auxiliary information on the presence and type of noise, quality of the samples, annotator confidence and a textual description, offering expert commentary and annotation rationale. Detailed information on all available labels is provided in the Supplemental Material.

## 4 Benchmark

To enable a fair comparison of future methods for InSAR based volcanic unrest detection, we provide the first benchmark on *Hephaestus Minicubes*. This benchmark is designed to serve as a strong baseline across two fundamental tasks: binary ground deformation classification and semantic segmentation. To transform our problem to a binary task, we group all sub-classes of ground deformation (*e.g.* Mogi, Sill) into one class. Below we present the main decisions made for the experimental setup. For detailed information on the complete experimental framework, readers are referred to the Supplemental Material.

**Data Split.** We apply a temporal data split, separating InSAR products by the primary acquisition date. The training set includes interferograms with the primary SAR acquired between January 1, 2014, and May 31, 2019, while the validation from June 1, 2019, to December 31, 2019. Finally the test set consists of interferograms with primary acquisition dates between January 1, 2020, and December 31, 2021. This split was carefully chosen to maintain spatial diversity by including data from all the available frames, with an adequate ratio of positive samples in each set, as seen in Tab. 2.

**Data Preparation.** To reduce input size, each InSAR product is cropped to $512 \times 512$ pixels. We apply cropping with a random offset from the frame center, ensuring any existing deformation patterns are included within the cropped area.We address class imbalance, by undersampling during training, using all available positive samples and an equal number of random negative samples in each epoch, and apply data augmentation to improve model generalization.

**Constructing InSAR Time-Series.** Constructing meaningful InSAR time-series is non-trivial due to the bi-temporal nature of each product, characterized by both primary and secondary acquisition dates. Moreover, the temporal gap between the primary and secondary acquisition in *Hephaestus* is not fixed, making temporal ordering highly ambiguous. In our framework, we define a valid time series as a sequence of interferograms that share the same primary and different secondary acquisition

Table 2: Summary of the temporal split windows and class distribution for both the single-timestep and time-series approaches.

| Split | Dates | Single-Timestep | | Time-Series | |
|---|---|---|---|---|---|
| | | Positives | Negatives | Positives | Negatives |
| Training | Jan 2014 – May 2019 | 1143 | 8697 | 701 | 2626 |
| Validation | Jun 2019 – Dec 2019 | 154 | 2416 | 75 | 728 |
| Test | Jan 2020 – Dec 2021 | 509 | 5992 | 225 | 1776 |
| Sum | Jan 2014 – Dec 2021 | 1806 | 17105 | 1001 | 5130 |

dates. These sequences are then ordered chronologically based on the secondary dates, as illustrated in Fig. 3.

This formulation allows models to observe the evolution of deformation relative to a fixed reference, providing insights into the dynamics of volcanic unrest. At the same time, sampling different secondary acquisitions can expose the model to variations in atmospheric noise, thereby encouraging learning of more robust, discriminative features [Dzurisin, 2003].

The number of valid InSAR products per primary SAR acquisition date varies across the dataset. To maintain a consistent input shape for model training, we either select all subsets that match the target sequence length or apply controlled duplication of available products when necessary. After examining the distribution of available secondary dates for each primary date, we choose to construct *time-series of length 3*, aiming for a balance between a rich temporal sequence and limited duplications.

In the time-series setting, labels are aggregated across the sequence. For the classification task, a sequence is considered positive if at least one of the products is labeled as showing deformation. For the segmentation task, the target mask is defined as the union of all individual deformation masks across the sequence. This approach ensures that models can leverage temporal information while maintaining a single target.

**Models.** We employ a diverse set of state-of-the-art models, widely used in Earth observation benchmarks (*e.g.* GEO-Bench Lacoste et al. [2023]), assessing their capacity for ground deformation classification and segmentation. For the classification task, we include ResNet-50 [He et al., 2016], Vision Transformer (ViT) [Dosovitskiy et al., 2020], ConvNeXt [Liu et al., 2022], MobileNetV3 [Howard et al., 2019], and EfficientNetV2 [Tan and Le, 2021], all pretrained on ImageNet [Deng et al., 2009]. For the segmentation task, we use UNet [Ronneberger et al., 2015], DeepLabv3 [Chen et al., 2017] and SegFormer [Xie et al., 2021], with ResNet-50 backbones pretrained on ImageNet.

**Evaluating Input Contributions.** Exploiting the diverse information of *Hephaestus Minicubes*, we examine the models' performance across varying configurations to evaluate the importance of each available data source. We vary the input on two dimensions for both classification and segmentation tasks. First, we examine the significance of temporal context in detecting volcanic unrest on both single-timestep and time-series setups. Second, we assess the impact of the auxiliary atmospheric variables by evaluating the models' performance with and without them. In Tabs. 3 and 4, we present the classification and segmentation results, respectively, reporting Precision, Recall, F1-score, and Area Under the Receiver Operating Characteristic curve (AUROC) for the classification task, and Precision, Recall, F1-score, and Intersection over Union (IoU) for the segmentation task. To account for variability introduced by initialization, undersampling, and augmentations, we report the average performance along with standard deviation over three random seeds.

## 5 Discussion

**Overall performance.** Examining the performance of the classification baselines in Tab. 3, we observe strong discriminative capability reaching up to $\approx 79\%$ in F1-Score. However, this performance declines in the segmentation task reaching up to $\approx 71\%$. This is not a surprising behavior, as exact delineation of ground deformation is often non-trivial even for experts. Even after discerning true ground deformation fringes from atmospheric contributions, defining the extent of such fringes is an ambiguous process, especially in regions with high incoherence. Such noise is inherent to the data

Table 3: Deformation classification metrics (mean ± std) for best model configurations between different random seeds. The tables report Precision (Prec), Recall (Rec), F1-score (F1), and Area Under the Receiver Operating Characteristic curve (AUROC) for the deformation class. The best value in each column is marked in **bold**, and the second best is underlined.

| | Model | Atm. | Prec | Rec | F1 | AUROC |
|---|---|---|---|---|---|---|
| **Single-Timestep** | ResNet-50 | ✗ | 83.63 ± 2.94 | 68.5 ± 3.05 | 75.29 ± 2.74 | **96.99** ± 0.39 |
| | | ✓ | 87.53 ± 3.7 | 64.37 ± 1.64 | 74.18 ± 2.32 | 96.26 ± 0.88 |
| | MobileNetV3 | ✗ | **95.03** ± 1.17 | 64.77 ± 0.88 | 77.02 ± 0.37 | 92.06 ± 2.61 |
| | | ✓ | 89.56 ± 0.54 | 69.02 ± 1.79 | 78.06 ± 1.13 | 91.99 ± 2.32 |
| | EfficientNetV2 | ✗ | 76.28 ± 3.74 | 44.66 ± 2.71 | 56.18 ± 1.08 | 74.98 ± 4.13 |
| | | ✓ | 82.28 ± 3.55 | 49.44 ± 4.22 | 61.61 ± 3.1 | 83.25 ± 5.02 |
| | ConvNeXt | ✗ | 93.04 ± 1.85 | **69.09** ± 2.01 | 79.25 ± 0.69 | 90.01 ± 3.88 |
| | | ✓ | 93.58 ± 0.3 | 68.76 ± 1.89 | **79.26** ± 1.33 | 90.29 ± 1.75 |
| | ViT | ✗ | 85.16 ± 11.21 | 55.8 ± 10.03 | 67.3 ± 10.51 | 87.58 ± 5.02 |
| | | ✓ | 90.75 ± 1.51 | 59.27 ± 7.47 | 71.45 ± 5.45 | 88.6 ± 3.0 |
| **Time-Series** | ResNet-50 | ✗ | 67.79 ± 2.11 | 60.0 ± 0.36 | 63.64 ± 0.89 | **92.68** ± 1.84 |
| | | ✓ | 68.65 ± 0.85 | 59.41 ± 3.27 | 63.66 ± 2.08 | 88.0 ± 2.33 |
| | MobileNetV3 | ✗ | 64.08 ± 1.38 | 63.56 ± 3.82 | 63.79 ± 2.54 | 89.39 ± 1.16 |
| | | ✓ | 63.51 ± 4.46 | 65.48 ± 5.88 | 64.29 ± 3.71 | 89.29 ± 1.06 |
| | EfficientNetV2 | ✗ | 68.58 ± 2.94 | 49.04 ± 7.26 | 56.88 ± 5.36 | 82.22 ± 3.5 |
| | | ✓ | 64.23 ± 9.07 | 56.0 ± 2.97 | 59.42 ± 4.2 | 82.63 ± 1.54 |
| | ConvNeXt | ✗ | 73.19 ± 2.02 | **68.89** ± 15.12 | **70.36** ± 8.63 | 91.65 ± 5.01 |
| | | ✓ | 75.88 ± 7.14 | 57.48 ± 2.55 | 65.36 ± 4.3 | 78.24 ± 3.18 |
| | ViT | ✗ | **80.52** ± 5.61 | 53.48 ± 4.62 | 63.92 ± 1.69 | 91.21 ± 3.26 |
| | | ✓ | 71.19 ± 2.74 | 61.63 ± 13.4 | 65.54 ± 8.5 | 89.2 ± 3.42 |

Table 4: Deformation segmentation metrics (mean ± std) for best model configurations between different random seeds. The tables report Precision (Prec), Recall (Rec), F1-score (F1), and Intersection over Union (IoU) for the deformation class. The best value in each column is marked in **bold**, and the second best is underlined.

| | Model | Atm. | Prec | Rec | F1 | IoU |
|---|---|---|---|---|---|---|
| **Single-Timestep** | DeepLabv3 | ✗ | **81.41** ± 1.42 | **63.74** ± 0.52 | **71.49** ± 0.30 | **55.63** ± 0.37 |
| | | ✓ | 81.64 ± 1.77 | 60.82 ± 2.61 | 69.65 ± 1.48 | 53.46 ± 1.75 |
| | UNet | ✗ | 82.43 ± 0.64 | 61.25 ± 1.27 | 70.27 ± 0.63 | 54.17 ± 0.75 |
| | | ✓ | 81.70 ± 0.41 | 53.71 ± 1.89 | 64.80 ± 1.48 | 47.95 ± 1.62 |
| | SegFormer | ✗ | 80.87 ± 1.74 | 61.32 ± 1.96 | 69.70 ± 0.70 | 53.50 ± 0.82 |
| | | ✓ | 83.13 ± 2.30 | 55.71 ± 0.98 | 66.68 ± 0.22 | 50.01 ± 0.25 |
| **Time-Series** | DeepLabv3 | ✗ | 75.68 ± 2.15 | 54.66 ± 1.54 | 63.42 ± 0.25 | 46.44 ± 0.27 |
| | | ✓ | 74.64 ± 2.48 | 46.67 ± 2.15 | 57.39 ± 1.84 | 40.27 ± 1.82 |
| | UNet | ✗ | 74.57 ± 3.10 | **58.57** ± 2.39 | 65.50 ± 0.35 | 48.7 ± 0.39 |
| | | ✓ | 70.28 ± 4.56 | 44.18 ± 1.64 | 54.19 ± 2.03 | 37.2 ± 1.92 |
| | SegFormer | ✗ | **79.22** ± 0.55 | 57.87 ± 1.46 | **66.87** ± 0.77 | **50.23** ± 0.87 |
| | | ✓ | 77.14 ± 0.10 | 47.88 ± 2.73 | 59.05 ± 2.10 | 41.92 ± 2.12 |

itself, making the annotation, and thereby accurate prediction challenging Kondylatos et al. [2025]. Many works have sought to improve segmentation capabilities in such conditions *e.g.* Acuna et al. [2019], Yu et al. [2018b]. Our benchmark establishes a strong reference point for future methods aimed at addressing these complexities.

**Impact of temporal dimension.** Examining both classification and segmentation experiments, we note a surprising drop in performance when we use a time-series input. While the theoretical advantages of using time-series data to capture volcanic unrest and account for atmospheric delays

are well established [Dzurisin, 2003], it is important to note that performance comparisons between single-timestep and time-series inputs are not entirely equivalent in our framework. Although both approaches aim to detect the same underlying geophysical phenomena, they operate on different data subsets due to the stricter requirements for constructing valid time-series (See Tab. 2). More importantly, the task formulation shifts: single-timestep models predict deformation masks for individual images, while time-series models segment the union of deformation patterns across multiple observations, effectively capturing the total extent of the affected area. As such, while performance trends are informative, variability in absolute metrics between the two setups should be interpreted with these structural differences in mind.

**Impact of atmospheric variables.** The impact of atmospheric information varies across tasks. In classification, some models exhibit modest performance gains with the inclusion of atmospheric context, with EfficientNetV2 and ViT demonstrating a consistent improvement both in the single-timestep and the time-series settings. In contrast, atmospheric information does not lead to improved performance in segmentation models. This discrepancy likely reflects the different demands of the two tasks. While atmospheric variables offer valuable insights into the atmospheric conditions, they are available at a substantially coarser spatial resolution than the InSAR data itself. This resolution mismatch may constrain their effectiveness, particularly in segmentation tasks where fine-grained spatial detail is critical for distinguishing true deformation from confounding patterns. Moreover, InSAR data offer significantly stronger information for the detailed delineation of ground deformation, which may lead it to dominate the learning process diminishing the influence of atmospheric input.

Motivated by this discussion, we examine specific cases where the inclusion of atmospheric variables helps the model mitigate false positives caused by atmospheric delays in semantic segmentation. In doing so, we aim to explore potential nuances that are not fully captured by aggregate performance metrics, in order to better understand the limitations of our baseline modeling approach and identify possible paths forward. In Fig. 4 we compare predicted masks from models trained with and without atmospheric inputs. Along with this, we also investigate the mean lateral gradient of total column water vapour (TCWV) across the primary and secondary SAR acquisition dates, contextualized against the broader distribution of mean TCWV gradients for the given frame, as lateral variation in atmospheric moisture is a key driver of atmospheric phase delays in InSAR measurements. Notably, both scenes exhibit high lateral variation in TCWV, suggesting that the atmospheric component can, under specific conditions, contribute meaningful information about phase delay artifacts. However, incorporating this knowledge into deep learning models remains a non-trivial challenge, particularly due to the aforementioned mismatch in resolution and information density between atmospheric inputs and InSAR data.

Finally, the inclusion of the atmospheric component as additional channels, for both primary and secondary SAR acquisitions, comes with a significant increase in input dimensionality, thereby adding to model complexity and potentially hindering performance. Effective and efficient handling of the atmospheric component remains a non-trivial and open challenge. Addressing this issue may require more sophisticated and context-aware approaches that model both the internal relationships of the atmospheric variables, as well as cross-modal interactions with the InSAR imagery.

## 5.1 Limitations

Despite extensive efforts in the annotation process, which incorporates validation from both internal and external sources, annotating InSAR imagery remains inherently challenging. The labels used for training and evaluation are not free from noise, reflecting the complexities involved in detecting volcanic unrest. The subtlety and variability of volcanic deformation patterns often lead to ambiguities in interpretation, making accurate and consistent annotation difficult, especially in regions with low signal coherence. This limitation is compounded by the fact that some volcanic events are subtle or evolve over extended periods, which can further complicate the identification and classification of deformation signals in the data.

Additionally, the temporal scale and nature of volcanic activity introduce significant challenges. Some unrest episodes are subtle and unfold over many years, while others are abrupt and short-lived. The frequency and expression of these events vary widely between volcanoes; some may remain inactive for extended periods before suddenly exhibiting signs of unrest. Consequently, certain volcanoes may not show any positive samples during the dataset's timeframe, limiting the model's exposure to their activity patterns. As a result, models trained under such constraints may struggle to generalize

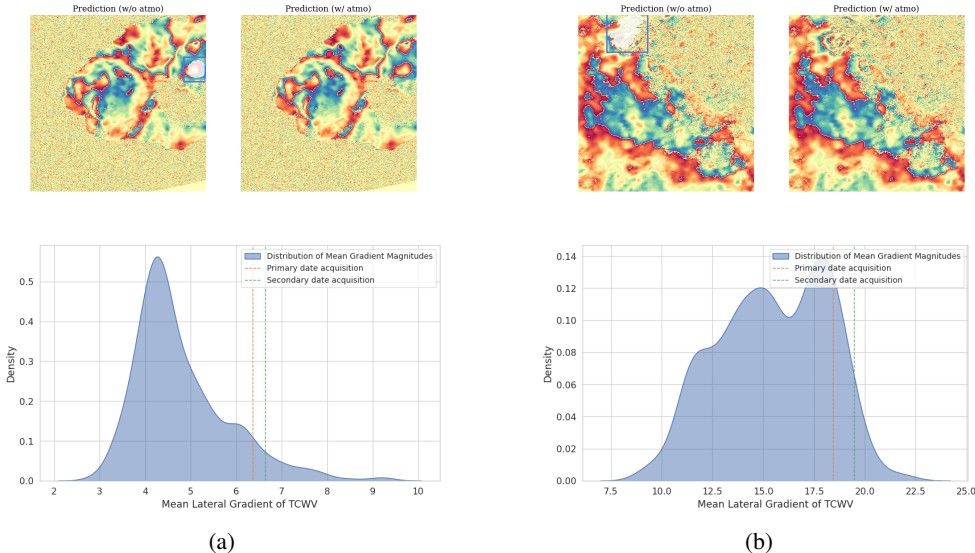

Figure 4: Compact view of DeepLabV3 predictions (top) with and without atmospheric input and the associated mean lateral gradient distributions of TCWV (bottom) for two samples: (a) Sierra Negra volcano, Galápagos islands. (15/3/2020 - 15/04/2020), (b) Valle de Piedras Encimadas region in Puebla, Mexico (05-08-2020 - 17/08/2020). We examine, representative examples where the inclusion of atmospheric variables leads to improved segmentation performance by mitigating false positives linked to atmospheric artifacts. In both cases, this improvement coincides with high lateral variation in TCWV, hinting at the potential value of atmospheric variables.

effectively in operational settings, particularly when tasked with detecting unrest at volcanoes with sparse or no prior positive observations.

# 6    Conclusion

*Hephaestus Minicubes* represents a significant advancement in data-driven volcanic unrest monitoring. By integrating high-resolution InSAR phase and coherence products, digital elevation models, atmospheric information and expert annotations into structured spatiotemporal datacubes, the dataset provides a rich foundation for machine learning research in this domain. The additional inclusion of atmospheric variables addresses a key challenge in InSAR analysis—distinguishing true ground deformation from atmospheric phase delays that can mimic similar patterns.

Building on *Hephaestus Minicubes*, we provide an extensive benchmark demonstrating the dataset's ability to support volcanic unrest monitoring as a multi-modal, multi-temporal problem. We examine two fundamental tasks, *i.e.* InSAR classification and semantic segmentation, across both single-timestep and time-series formats. Our results establish strong baselines for future applications, assessing the capacity of state-of-the-art architectures in this domain. The inclusion of time-series and atmospheric data—while theoretically promising—reveals practical complexities. Similarly, while atmospheric variables can help mitigate false positives from phase delay artifacts, their coarse spatial resolution and increased model complexity limit their overall utility in segmentation tasks. These findings underscore the importance of developing more nuanced, context-aware modeling strategies that can effectively leverage atmospheric information and temporal context, pointing to promising directions for future research.

*Hephaestus Minicubes* is, to the best of our knowledge, the first large-scale machine learning ready dataset to incorporate such diverse information. We believe *Hephaestus Minicubes* will be a valuable asset to the research community, contributing to the growing integration of data-driven methods within Earth science applications.

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
