# Hephaestus Minicubes:
# A Global, Multi-Modal Dataset for Volcanic Activity Monitoring

**Nikolas Papadopoulos** [1]
npapadopoulos@mail.ntua.gr

**Nikolaos Ioannis Bountos**[1,2]
bountos@noa.gr

**Maria Sdraka**[1,2]
masdra@noa.gr

**Andreas Karavias**[1]
andreas_karavias@mail.ntua.gr

**Ioannis Papoutsis**[1]
ipapoutsis@mail.ntua.gr

[1] **Orion Lab**
National Observatory of Athens & National Technical University of Athens

[2] **Harokopio University of Athens**

# Supplemental material

## A    Detailed Dataset Description

This section details the annotation schema used in *Hephaestus Minicubes*. Building upon the expert labels from the original *Hephaestus* dataset [Bountos et al., 2022], annotations were adapted to a spatiotemporal format compatible with the datacube structure. A comprehensive view of the dataset variables can be seen in Tab. 1 and examples of different annotation types are shown in A.1. In the following paragraphs we will detail the key variables of *Hephaestus Minicubes*.

**Activity.**    Activity-related segmentation masks lie at the core of the dataset's annotation schema detailing the presence of deformation, its geophysical source and its intensity.

A high-level *Deformation Mask*, delineates `Volcanic` and `Earthquake` induced deformation patterns.

The *Activity Type Mask* captures more detailed geophysical context, distinguishing among several common deformation source models: `Mogi` [Kiyoo, 1958], `Dyke` [Okada, 1985, Sigmundsson et al., 2010], `Sill` [Fialko et al., 2001], and `Spheroid` [Yang et al., 1988], as well as deformation attributed to `Earthquake` events or labeled as `Unidentified` when no clear model can be observed.

The *Intensity Level Mask* categorizes the strength of deformation signals based on the number of visible fringes in the interferogram: `Low` (1 fringe), `Medium` (2–3 fringes), and `High` (more than 3 fringes). For earthquake-related events, the intensity is marked as `None`, reflecting their distinct deformation characteristics.

Additionally, we include a *Phase* categorical variable that captures the state of the volcano *i.e.* `Rest` (no sign of volcanic activity), `Unrest` (indicating uplift), or `Rebound` (indicating subsidence). Again, for Earthquake events, we set the phase to `None`.

**Noise.**    A separate set of annotation variables aim to capture specific signal characteristics and noise patterns. These include: *glacier fringes*, when observed fringes result from glacier melting (`1` if present, `0` otherwise); *orbital fringes*, for phase ramps caused by satellite orbital errors; and *atmospheric fringes*, which take four values: `type 0` (no atmospheric impact), `type 1` (vertical

Table 1: Overview of Hephaestus Minicubes Dataset Variables

| Variable | Description |
| --- | --- |
| **InSAR Products** | |
| Phase Difference | Difference in SAR signal phase, indicating surface displacement. |
| Coherence | The reliability of the interferometric phase measurement. |
| **Topography** | |
| DEM | Digital Elevation Model: a representation of Earth's surface topography. |
| **Atmospheric Variables** | |
| Total Column Water Vapour | Water vapour from surface to the top of the atmosphere. |
| Surface Pressure | Atmospheric pressure at Earth's surface. |
| Vertical Integral of Temperature | Mass-weighted temperature integral from surface to top of the atmosphere. |
| **Annotations** | |
| Deformation Mask | High level mask for deformation presence {`Non-Deformation`, `Volcanic`, `Earthquake`}. |
| Activity Type Mask | Mask identifying the activity type: {`Sill`, `Dyke`, `Mogi`, `Spheroid`, `Earthquake`, `Unidentified`}. |
| Intensity Level Mask | Mask identifying the intensity level of the activity {`None`, `Low`, `Medium`, `High`}. |
| Phase | Phase of the activity: {`Rest`, `Unrest`, `Rebound`, `None` }. |
| Atmospheric Fringes | Identifies types of atmospheric noise. |
| Glacier Fringes | Identifies deformation patterns from glacier melting. |
| Orbital Fringes | Identifies phase ramps due to orbital errors. |
| Corrupted | Flag for corrupted data. |
| No Info | Identifies low-coherence interferograms, where meaningful interpretation is not possible. |
| Low Coherence | Identifies samples that are characterized from interferometric signal decorrelation. |
| Is Crowd | Identifies whether multiple deformation masks exist. |
| Caption | Expert text description of the interferogram and annotation rationale. |
| Confidence | Value indicating the annotator's confidence [`0-1`]. |
| **Metadata** | |
| Unique Id | Unique identifier for each sample. |
| Valid Date Pair | Boolean flag for primary/secondary dates with existing InSAR products. |

stratification correlated with topography due to changes in the troposphere's refractive index), `type 2` (turbulent mixing and vapors caused by liquid or solid particles in the atmosphere), and `type 3` (a combination of `type 1` and `type 2`). The *low coherence* variable is set to `1` when interferometric signal decorrelation dominates the image. *No info* is used when coherence is so low throughout the interferogram that meaningful interpretation is not possible.

**InSAR Processing Errors.** Automated InSAR generation pipelines may, occasionally, result in faulty products. To identify corrupted InSAR, and facilitate automatic detection of such instances in future applications, we include a set of annotation variables denoting technical faults. The first is the binary variable *corrupted*, which identifies interferograms that are entirely problematic and unusable. The second is *processing error*, which distinguishes between specific InSAR processing errors: `type 1` refers to debursting errors during the synchronization of bursts from one or more sub-swaths; `type 2` indicates Sentinel-1 sub-swath merging errors, which appear as visible discontinuities, while `type 0` denotes interferograms free of such processing issues.

**Meta-Information.** The *confidence* score is a continuous value in the range [`0, 1`] that reflects the annotator's confidence regarding the deformation classification. The *is crowd* variable is set to `0` when at most one local fringe pattern is present and `1` when two or more such patterns appear within the same interferogram. Furthermore, the *caption* field contains a text description providing expert commentary, interpretation rationale, or relevant contextual notes for the InSAR phase difference product. Finally the *unique id* serves as a unique identifier for each sample, and the *valid date pair* flag indicates whether the combination of primary and secondary dates corresponds to an existing InSAR product, intended for use in the multi-dimensional array functionality.

**Annotation Process.** The annotation process was carried out by a team of InSAR experts through photo-interpretation on the available wrapped interferograms. Each image was optically inspected over the locations of volcanoes, as well as the surrounding areas, in order to evaluate the quality of the interferograms and indicate the impact of atmospheric delays and potential artifacts. Each

positive annotation was cross-validated via external sources (publications, reliable news sources) as well as the COMET Volcano Deformation Portal [1], to support the evaluation of whether the observed displacement was due to true volcanic activity or atmospheric delay effects. For further details regarding the annotation process readers are referred to the original *Hephaestus* dataset [Bountos et al., 2022].

## A.1 Visualizing Annotation Diversity

In this section we present selected examples of different annotation variables, illustrating the diversity and richness of the annotation variables in *Hephaestus Minicubes*.

**Text Captions.** In Fig. 1, we showcase InSAR imagery with various apparent deformation patterns and atmospheric phenomena, together with the expert textual captions, highlighting the complexity and variability captured in the dataset. Each caption provides a thorough description of the location and type of all underlying phenomena, as well as a reference to interpretation challenges if present.

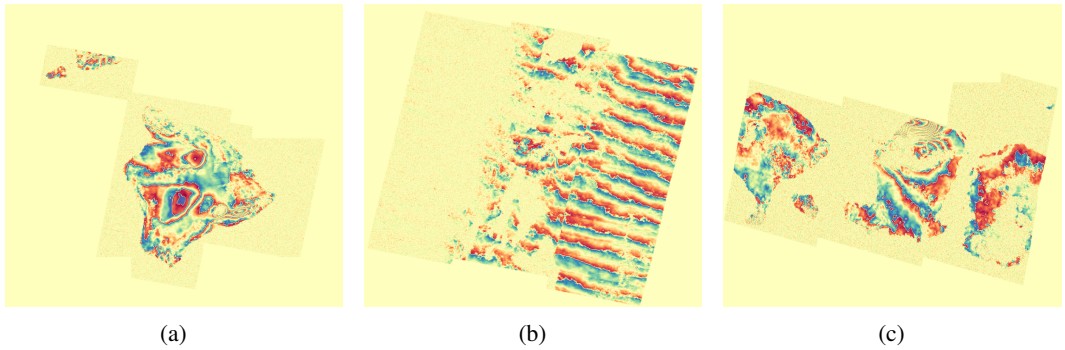

(a)  (b)  (c)

(a) "Vertical stratification can be detected on the high altitude areas. Turbulent mixing effect can also be detected on the right, central and top-left side. Two deformation patterns can be detected on the right side. A dyke-type of high intensity on the leftmost and sill-type of medium intensity on the rightmost. "
(b) "Turbulent mixing effect or wave-like patterns caused by liquid and solid particles of the atmosphere can be detected around the area. No deformation activity can be detected. Orbital fringes detected. Difficulties in extracting information."
(c) "Noise can be detected on the bottom-right area. Turbulent mixing effect can be detected on the wider left and central side of the region. Vertical stratification can also be detected on the central-top side of the region. An earthquake deformation pattern can be detected on the top-central side of the region."

Figure 1: Textual annotations highlighting volcanic and atmospheric phenomena in InSAR imagery from the *Hephaestus Minicubes* dataset.

**Activity and Intensity Masks.** In Fig. 2, we present different time-series of InSAR products overlaid with expert-provided *activity type* and *intensity level* masks. These examples illustrate the temporal evolution of deformation signals and how distinct geophysical phenomena are annotated both categorically and in terms of intensity, emphasizing the structured labeling within *Hephaestus Minicubes*. The InSAR products are shown in grayscale for visual clarity.

## B Extended Experimental Details

In this section we provide additional details on the experimental configurations used in the provided benchmark.

**Time-series Construction.** We construct the time-series by first grouping samples based on their primary SAR acquisition date. The distribution of the length of the resulting time-series can be seen in Fig. 3. Based on this, we fix the time-series length to 3, to ensure input consistency. For primary SAR acquisitions that produce time-series with length greater than 3, we extract all possible sub-sequences of the predefined length. Conversely, if they contain fewer than 3, we randomly duplicate available interferograms to reach the required time-series length.

---

[1] https://comet.nerc.ac.uk/comet-volcano-portal/

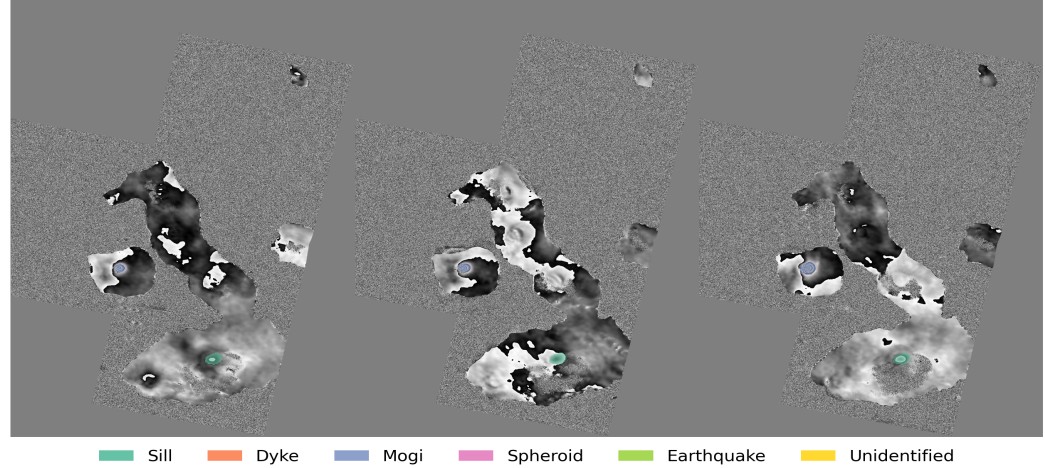

(a) Overlay of *activity type* masks on a time-series of InSAR phase difference products.

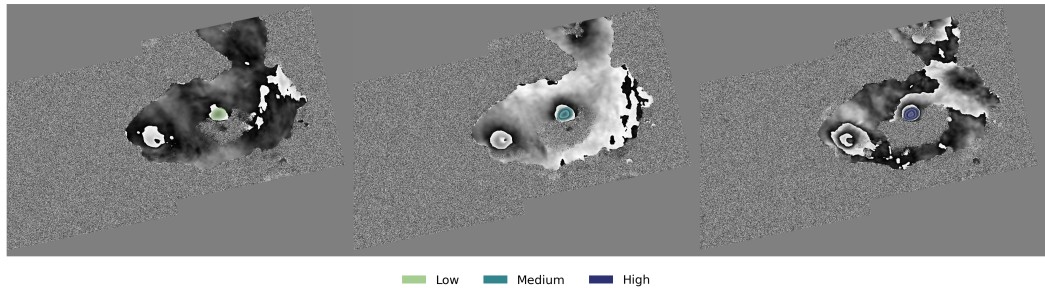

(b) Overlay of *intensity level* masks on a time-series of InSAR phase difference products.

Figure 2: Examples of annotated InSAR products with overlayed annotated masks indicating: (a) distinct geophysical activity types and (b) activity intensity levels.

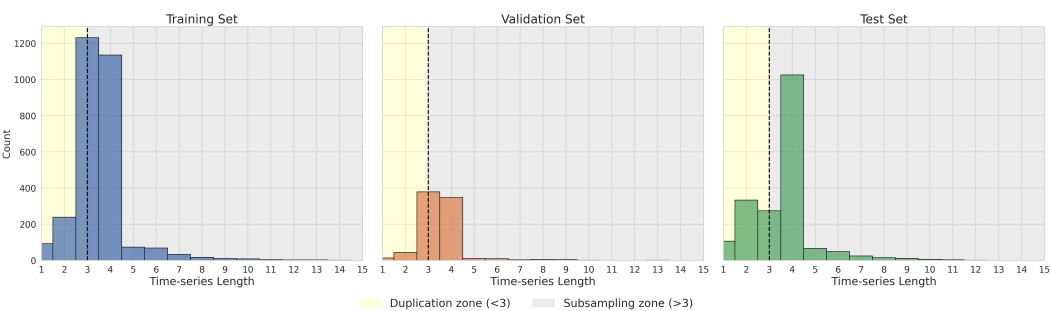

Figure 3: Distribution of time-series lengths across training, validation, and test sets.

**Training Setup.** We conducted all experiments on a single GPU (NVIDIA GeForce RTX 3090 Ti). We trained all models for 90 epochs, using the AdamW optimizer [Loshchilov and Hutter, 2017] with a fixed learning rate of $10^{-5}$ and a weight decay set to $10^{-2}$. For classification, we found that cross-entropy loss consistently delivered the best performance, except for ViT that displayed more stable convergence with focal loss. For segmentation, focal loss was more effective, likely due to its robustness to class imbalance in pixel-wise annotations [Lin et al., 2017]. To mitigate the effect of class imbalance, we employed an undersampling strategy during training. For each epoch, all available positive samples were included, along with a randomly selected subset of negative samples of equal size. All models were trained with a random set of data augmentations including gaussian blur, random resize crop, horizontal and vertical flips and random rotations. Tables 2 and 3 report the

number of trainable parameters for all models along with the average runtime of each experiment for both classification and segmentation tasks respectively.

Table 2: Overview of classification architectures

| Model architecture | # Trainable parameters (M) | | # Average runtime (hours) | |
|---|---|---|---|---|
| | Single-image | Time-series | Single-image | Time-series |
| ResNet-50 | 23.5 | 23.6 | 2.2 | 2.7 |
| MobileNetV3 (Large) | 4.2 | 4.2 | 1.7 | 2.0 |
| EfficientNetV2 (Small) | 20.2 | 20.2 | 2.6 | 2.7 |
| ConvNeXt (Small) | 87.6 | 87.6 | 4.9 | 5.4 |
| ViT (Small) | 22.3 | 24.0 | 2.5 | 3.2 |

Table 3: Overview of segmentation architectures

| Model architecture | Backbone | # Trainable parameters (M) | | # Average runtime (hours) | |
|---|---|---|---|---|---|
| | | Single-image | Time-series | Single-image | Time-series |
| DeepLabV3 | ResNet-50 | 26.7 | 26.8 | 2.3 | 2.9 |
| UNet | ResNet-50 | 32.5 | 32.6 | 2.3 | 3.0 |
| SegFormer | ResNet-50 | 24.9 | 24.9 | 1.6 | 3.7 |

# C  Qualitative Results and Model Insights

In this section, we present indicative examples that provide insights on models' performance beyond quantitative metrics, especially given the inherent ambiguities and complexities of the data.

**Ambiguity in Segmentation Masks.**  As discussed in section 5 of the main text, delineating the exact boundaries of deformation is often ambiguous, especially in regions of high-incoherence. Such noise is inherent to the data itself, making the annotation, and thereby accurate prediction challenging. In Fig. 4 we offer two representative examples of qualitatively good model predictions that do not perfectly align with the annotator's estimation, indicating the aforementioned ambiguity. In this case, model predictions accurately identify the observable deformation patterns to some extent, but exclude the noisy areas. The human annotator, however, includes these noisy regions, considering them as part of the event, even if they do not exhibit ground deformation fringes.

**Time-Series Predictions.**  Fig. 5 illustrates important insights concerning model predictions on time-series inputs. Although both time-series of the figure focus on the same volcanic region, variations in the spatial extent or deformation onset across time-steps can affect detection ability. In the first sequence, less-pronounced deformation patterns are only visible in the final time-step, which the model fails to detect. In contrast, the second sequence displays a progressively intensifying deformation signal starting from the second time-step, providing stronger temporal cues that allow the model to correctly identify and segment the affected area. This example highlights how both the timing and strength of deformation signals across the time-series could play an important role in the the model's ability to identify deformation.

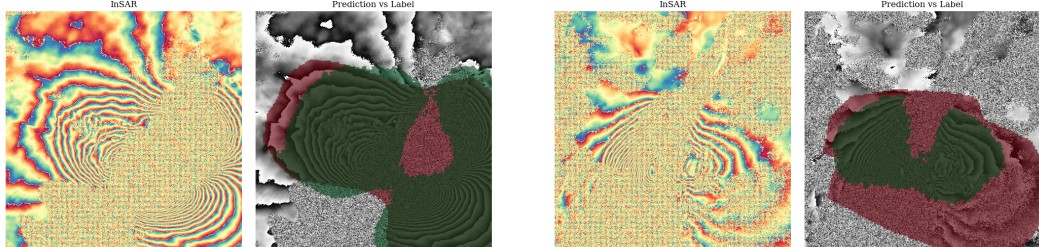

Figure 4: Qualitative examples of true positive model predictions that illustrate the inherent ambiguity in segmentation masks, where deformation extent is difficult to define. The left image of each pair depicts the InSAR phase difference, while the right image overlays the ground truth mask (in red) and the model prediction (in green) on the InSAR product (in grayscale). The results presented here have been obtained by the best performing DeepLabV3 model, which utilizes atmospheric data as input.

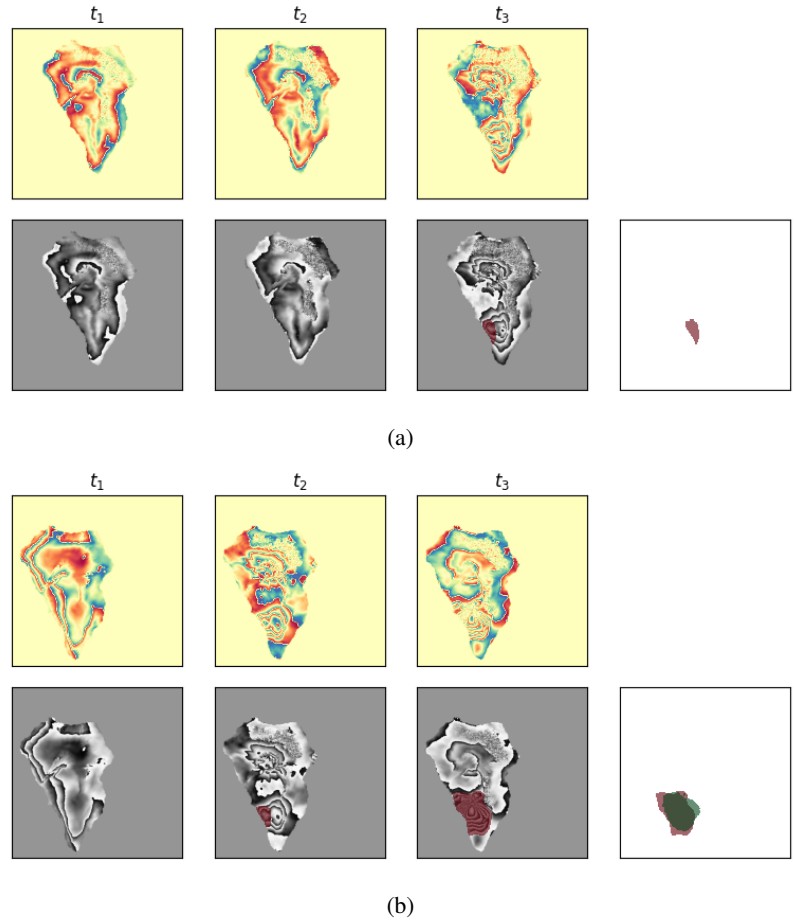

Figure 5: Time-series examples over La Palma, Tenerife, demonstrating the role of temporal context in model predictions. The upper image series of each subfigure depicts the InSAR products in each timestep, and the lower image series depicts the ground truth masks (in red) for each time step with the corresponding InSAR in grayscale as the background. The last image in the lower series depicts the union of all ground truth masks (in red) and the model predictions (in green). Results have been obtained with the best performing SegFormer model (with atmospheric input). Subfigure a) shows a false negative case where only the final time-step exhibits deformation, which the model fails to detect, while subfigure b) illustrates a case where the model successfully identifies the deformation due to its increased presence in the last two time-steps. These examples highlight how variations in signal strength and onset time can influence model prediction.