# OpenReview forum: "Hephaestus Minicubes: A Global, Multi-Modal Dataset for Volcanic Unrest Monitoring"
_NeurIPS.cc/2025/Datasets_and_Benchmarks_Track — Submitted to NeurIPS 2025 Datasets and Benchmarks Track_

### Official Review · Reviewer_VjjP · 2025-07-01

**Rating:** 4
**Confidence:** 2

**Summary:**

The work introduces Hephaestus Minicubes, a global collection of 38 spatiotemporal datacubes that cover 44 of the world's most active volcanoes over a 7-year period (2014-2021). Each spatiotemporal datacube integrates InSAR products, topographic data, and atmospheric variables known to affect SAR signal propagation. The dataset also includes expert annotations detailing deformation events' type, intensity, and spatial extent.

**Dataset Code Accessibility:**

Yes

**Dataset Code Comments:**

The author has released the dataset.

**Ethical Considerations:**

No, there are no or only very minor ethics concerns

**Final Justification:**

Thanks for the authors' response. I keep my positive rating.

**Limitations Weaknesses:**

1. Surprisingly, incorporating temporal sequences reduced performance in Table 3 and 4. While partly justified, this result warrants deeper analysis, including how temporal ambiguities impact learning effectiveness.
2. Given volcano-specific activity variability, the generalizability of learned models to unseen volcanoes or rare deformation types remains insufficiently discussed or evaluated.

**Strengths Contributions:**

1. This work addresses a significant gap in the availability of curated datasets for training deep learning models on InSAR data, which is crucial for volcanic unrest monitoring.
2. By integrating multiple sources of information, including InSAR phase difference, coherence, DEMs, and atmospheric variables, the dataset supports complex multi-modal tasks.
3. The authors establish a benchmark for evaluating state-of-the-art architectures on both binary ground deformation classification and semantic segmentation tasks, setting strong baselines.

---

> ### Author Rebuttal · Authors · 2025-07-30
>
> We thank the reviewer for the valuable feedback and comments. We will address each point individually in the order they were presented.
>
> ## Time-series Performance
>
> We agree that the performance drop when incorporating temporal sequences, as observed in Tables 3 and 4, is a particularly interesting result. In Section 5, we discussed some challenges on the comparison of the single-timestep and time-series approaches. Beyond that, several interrelated factors contribute to this result and warrant further consideration.
>
> First, as shown in Table 2, constructing time-series inputs significantly reduces the number of available training samples due to the requirement for consecutive, temporally aligned frames. This reduction may limit the model’s ability to generalize, especially given the natural scarcity of positive instances.
>
> Second, the reduced number of samples combined with the increased input dimensionality from stacking multiple timesteps, which results in a parameter increase at the first layer of the model, adds complexity to the learning process and may potentially lead to overfitting.
>
> Third, the formulation of the target label in the time-series setting may introduce additional ambiguity, especially for the segmentation task. In this case, we treat the target as the union of the ground truth across all time-steps. As discussed in Section 5, defining the extent of fringes is an ambiguous process, due to noise inherent to the data (e.g. atmospheric contributions and  high incoherence). When multiple such noisy signals are aggregated across time, this ambiguity may be amplified, significantly affecting the learning process.
>
> A similar issue arises in the classification setting, where we label a time-series as positive if deformation is present in any one of the frames. In this case the model may be penalized for failing to detect signals that are temporally sparse or weak, even if most frames contain no deformation. This effect might be amplified by the naive stacking of input timesteps, which does not explicitly model the temporal dynamics.
>
> We acknowledge that there is significant room for improvement in how temporal information is modeled, especially given its intuitive importance. Our benchmark is intended to provide strong baselines for the investigation and development of future methods. However, it is worth noting that Hephaestus Minicubes is the first dataset that makes such an investigation possible.
>
> We will include this discussion in Section 5 (Discussion) and Section 6 (Limitations) for the camera-ready version.
>
> ## Generalization Ability to Unseen Volcanoes
>
> We adopted a temporal split in our benchmark to more closely reflect the conditions of real-world early warning systems. In operational settings, the volcanoes under observation and their locations are typically known in advance, allowing models to leverage historical events at those specific sites.
>
> That said, we fully agree that assessing how well models generalize to previously unseen volcanoes is an important consideration, both for understanding performance across diverse geological contexts and for enabling real world early warning systems to expand to new regions. To address this, we conducted a preliminary study to assess spatial generalization with the following experimental setup.
>
> We keep a temporal split for training and validation similar to the current benchmark, but hold out a subset of two **previously unseen volcanoes** specifically for the test set. These volcanoes were selected based on their activity frequency profiles, representative of frequent activity (0.55 positive/negative ratio) and infrequent activity (0.03). The final test set has an overall positive/negative ratio of 0.07, closely matching that of the original benchmark split (0.085), to ensure a fair basis for comparison. This experimental setup simulates a real-world scenario where the monitoring system is extended to new volcanoes, while effectively assessing the spatial generalizability of the existing benchmark.
>
> In this preliminary study, we evaluate a DeepLabV3 with a ResNet50 as backbone for segmentation tasks, and a ResNet50 for classification. Our initial results for single-timestep input are quite encouraging:
>
> - Classification: 87.5% F1 with atmospheric input /  86.98% F1 without atmospheric input
> - Segmentation: 75.22% F1 with atmospheric input / 74.55% F1 without atmospheric input
>
> While defining a representative spatial split is non-trivial due to the natural class imbalance across positive and negative instances as well as ground deformation types and event intensities, our initial results suggest that the models exhibit promising spatial generalization.
>
> We will extend this study, complementing it with the full range of experiments, along with an investigation of the impact of varying spatial splits (which affects the distribution of ground deformation types, event intensities, state of the volcano e.g rest, rebound, as well as positive/negative ratio). We will include this in the supplemental material and also discuss it in Section 5 of the main text.

---

### Official Review · Reviewer_bz94 · 2025-07-02

**Rating:** 4
**Confidence:** 3

**Summary:**

The paper introduces Hephaestus Minicubes, a novel dataset designed for volcanic unrest monitoring using Interferometric Synthetic Aperture Radar (InSAR) data. The dataset comprises 38 high-resolution spatiotemporal datacubes covering 44 active volcanoes over a 7-year period, integrating InSAR phase and coherence products, digital elevation models (DEMs), atmospheric variables, and expert annotations. These annotations include deformation type, intensity, spatial extent, and textual descriptions. The dataset builds on the earlier Hephaestus dataset by improving ground sampling distance, incorporating additional data sources, and supporting time-series analysis. The paper also provides a comprehensive benchmark for two tasks: binary ground deformation classification and semantic segmentation, using state-of-the-art deep learning models such as ResNet-50, Vision Transformer (ViT), and UNet. The benchmark evaluates the impact of temporal context and atmospheric variables, achieving strong baselines.

**Additional Feedback:**

1.Given the acknowledged challenges with annotation noise (Section 5.1), could you elaborate on the process used for expert annotations, including how disagreements among annotators were resolved? Did you quantify inter-annotator agreement?
2.The authors mainly discussed data up to 2021 in the paper. Have they considered or tested more recent data?
3.In Figure 4, you highlight cases where atmospheric variables help mitigate false positives. Could you provide quantitative insights (e.g., how often such improvements occur across the dataset) or discuss specific atmospheric conditions that make these variables more effective?

**Dataset Code Accessibility:**

Yes

**Ethical Considerations:**

No, there are no or only very minor ethics concerns

**Limitations Weaknesses:**

1.The temporal split (training: 2014–May 2019; validation: June–Dec 2019; test: 2020–2021) ensures diversity but may introduce challenges due to temporal variability in volcanic activity.

2.By addressing the challenge of distinguishing true deformation from atmospheric artifacts, the dataset has significant potential for advancing data-driven volcanic monitoring, with implications for hazard assessment and risk mitigation.

**Strengths Contributions:**

1.Hephaestus Minicubes is a significant advancement over the original Hephaestus dataset, offering a unique, machine-learning-ready collection of multi-modal, multi-temporal datacubes.

2.The dataset includes detailed expert annotations, enhancing its utility for classification, segmentation, and potentially other tasks like captioning or anomaly detection.

3.The paper establishes a strong baseline for two tasks (classification and segmentation) using a diverse set of state-of-the-art models.

---

> ### Author Rebuttal · Authors · 2025-07-30
>
> We thank the reviewer for the constructive feedback and suggestions. We will try to address each of the main points as well as the additional feedback individually and in the order they were presented.
>
> ## Choice of Temporal Split
> We adopt a temporal split to better reflect the conditions of real-world early warning systems. In operational settings, the set of the observed volcanoes and their locations are known, and as a result we can leverage past observations of unrest. Temporal dynamics indeed play a key role: volcanic deformation evolves over time, exhibiting distinct patterns during onset, peak, and deflation. In addition, sensor noise and atmospheric conditions vary over time, making each observation unique in the temporal dimension despite the common spatial characteristics. These are challenges that would exist in real-world systems. Our proposed split is an attempt to mimic realistic scenarios for the development of methods with direct impact.
>
> ## Deformation and Atmospheric Artifacts
> We thank the reviewer for recognizing the potential of developing methods able to address the critical challenge of distinguishing true deformation from atmospheric artifacts. This was a primary motivation for including ERA5-derived atmospheric variables and DEMs in the dataset. Their inclusion is intended to support research into understanding and mitigating atmospheric artifacts, enabling more robust and reliable volcanic unrest monitoring.
>
> ## Additional Feedback
>
> ### Annotation Process Background
>
> Four InSAR experts participated in the annotation process, evenly distributing the workload and cross-validating 10% of the samples of each other's annotations in a cyclical intra-validation process. Each expert had access to relevant scientific literature, volcano monitoring reports, and reliable news sources. Challenging cases identified individually by the annotators or as disagreement during the cross-validation process were resolved through group discussion to reach a consensus. Annotations include a “confidence” field, reflecting the annotator’s certainty in their assessment. For difficult cases, this value represents the confidence in the final verdict.
>
> ### Testing on Recent Data
>
> We agree that incorporating more data would be beneficial, mainly due to the scarcity of positive samples. However, this is costly, requiring expert photo-interpretation of thousands of InSAR frames annually, often yielding only a few hundred positive cases. Even producing a focused case study for a specific event or volcano requires substantial effort, as generating reliable segmentation masks demands careful analysis across a substantial amount of samples to draw meaningful conclusions about predictive performance. Extending the dataset to incorporate more years and locations remains a potential direction for future work. In the present work our focus was to build on Hephaestus and improve in terms of ground sampling distance, expand the included data sources to account for atmospheric phase delays, and improve the structure of the dataset for ML practitioners, while providing a robust testbed to evaluate related future methods.
>
> ### Qualitative Results
>
> As noted in Section 3.2, the inclusion of atmospheric variables is motivated by the physical understanding that atmospheric phase delays can mimic deformation signals in InSAR imagery [1-3]. Figure 4 was intended to offer qualitative insight behind the physical processes underlying the InSAR signal and motivate future research on how such variables might be used more effectively, despite the overall drop in quantitative segmentation performance. Quantifying this effect across the dataset is challenging for two main reasons:
>
> a) The atmospheric variables contributing to this effect are multiple (humidity, pressure, temperature) and are relevant for both the primary and the secondary SAR acquisition. Producing a reliable quantitative metric that is able to capture this interconnected multivariable effect is non-trivial.
>
> b) Other confounding factors may also contribute to variations in results, such as sensor noise, terrain characteristics, or even spurious associations learned by the model.
>
> Considering the above, rigorously attributing the contribution of specific modalities to the predictive skill of our models requires a formal and systematic investigation, potentially using explainability methods. We acknowledge this as an important research direction.
>
> We will more clearly clarify the motivation behind these qualitative results and emphasize the need for tailored methods to fuse atmospheric variables with InSAR imagery, incorporating physics-based knowledge in real-world applications.
>
> [1] Zebker, H. A., Rosen, P. A., and Hensley, S. (1997). Atmospheric effects in interferometric synthetic aperture radar surface deformation and topographic maps. Journal of geophysical research: solid earth, 102(B4):7547–7563.
>
> [2] Massonnet, D. and Feigl, K. L. (1998). Radar interferometry and its application to changes in the earth’s surface. Reviews of geophysics, 36(4):441–500.
>
> [3] Beauducel, F., Briole, P., and Froger, J.-L. (2000). Volcano-wide fringes in ers synthetic aperture radar interferograms of etna (1992–1998): Deformation or tropospheric effect? Journal of Geophysical Research: Solid Earth, 105(B7):16391–16402.

---

> > ### Comment · Reviewer_bz94 · 2025-08-06
> >
> > Thanks to the author for the response! It answers some of my questions, but given the limitations, I'll keep my score.

---

### Official Review · Reviewer_A5oJ · 2025-07-02

**Rating:** 4
**Confidence:** 2

**Summary:**

The paper introduces Hephaestus Minicubes, a multi-modal, spatiotemporal dataset for volcanic unrest monitoring using InSAR (Interferometric Synthetic Aperture Radar). The dataset includes high-resolution InSAR phase/coherence products, DEMs, atmospheric variables, and expert annotations covering 44 active volcanoes from 2014–2021. The authors benchmark classification and segmentation tasks using state-of-the-art models, evaluating the impact of temporal context and atmospheric data on performance.

**Dataset Code Accessibility:**

Yes

**Ethical Considerations:**

No, there are no or only very minor ethics concerns

**Limitations Weaknesses:**

1. Focuses on 44 volcanoes, potentially missing underrepresented eruption styles or regions.

2. In my opinion, controlled acquisition conditions (Sentinel-1) may not fully capture field variability (e.g., dense vegetation, steep terrain).

3. No runtime/GPU memory metrics provided for benchmarking.Could you provide it?

4. I must point out with regret that your paper has not been anonymized.

**Strengths Contributions:**

1. The writing of the paper is well-executed, and the content is easy to understand.

2. This paper has some innovation. It is first Large-Scale ML-Ready InSAR Dataset.

3.This dataset has great research value. Meanwhile, It includes expert annotations (deformation type, intensity, phase) and textual descriptions for interpretability.

---

> ### Author Rebuttal · Authors · 2025-07-30
>
> We thank the reviewer for the thoughtful review. We will try to address each point individually in the order they were presented.
>
> ## Focus on 44 Volcanoes
>
> Hephaestus Minicubes builds on and extends the Hephaestus dataset (Sec.3). The selection of volcanoes was grounded on documented unrest or eruption events captured through scientific literature, monitoring reports, or reliable news sources. This strategy ensured a set of volcanoes with relatively frequent activity. This is crucial given the rare nature of positive instances for natural hazards, and the narrow temporal window of Sentinel-1 operations (2014-present).
>
> In addition, we ensured **broad geographic diversity**, covering six out of seven continents (excluding Antarctica), and included a substantial number of volcanoes classified as active but not exhibiting unrest during the annotated period.
>
> It is worth noting that Hephaestus Minicubes constitutes the largest, manually annotated ML-ready dataset in the domain.
>
> While expanding coverage to include more regions and underrepresented deformation types would be valuable, mainly due to the scarcity of positive cases, such efforts are constrained by the high cost of expert photo-interpretation and detailed annotation across thousands of InSAR frames per year—often yielding only a few hundred positive instances. Reasonably, further extending the dataset remains a potential direction for improvement and is left as future work. We will clarify this in the main text for the camera ready version.
>
> ## Sentinel-1 and Field Variability
>
> As discussed in the introduction section, InSAR data are well-suited for detecting surface motion, making them an ideal tool for global volcano monitoring. To complement this information, Hephaestus Minicubes incorporates elevation information, which can identify steep terrains, and atmospheric data known to directly contribute to phase delays in the SAR signal (Sec. 3.2). Additional modalities such as optical and multispectral imagery, while essential for multiple tasks, e.g. vegetation density and land cover classification, are not directly relevant to volcanic deformation monitoring, and are thus omitted in this work.
>
> That said, the effects of vegetation and other land cover characteristics are indirectly captured through the InSAR coherence modality, which serves as a proxy and is included in our dataset. Expanding Hephaestus Minicubes with other data sources that could potentially be utilized to improve prediction to volcanic unrest detection or related tasks is an open problem and left as a future work.
>
> ## Runtime/GPU Memory Metrics
>
> In Tables 6 and 7 of the Supplemental Material, we report the number of trainable parameters and the average training time per model, along with the GPU used (NVIDIA GeForce RTX 3090 Ti). To further enhance this information, we propose to include memory consumption during training for a fixed batch size in the supplemental material for the camera-ready version.
>
> ## Anonymization Concerns
>
> As per the official NeurIPS 2025 Datasets & Benchmarks Track guidelines, the submission process for this track follows a single-blind review format by default, where author identities are known to the reviewers.

---

> > ### Comment · Reviewer_A5oJ · 2025-08-05
> >
> > I sincerely thank the authors for their efforts in the research field, which undoubtedly has enhanced the overall level of the research. Given that I am not particularly proficient in this field, I will maintain the original score and reduce the confidence level.

---

### Official Review · Reviewer_Tq9w · 2025-07-03

**Rating:** 4
**Confidence:** 4

**Summary:**

The authors present Hephaestus Minicubes, a publicly available benchmark for global volcanic unrest monitoring. The dataset contains 38 spatiotemporal cubes (2014–2021, ≈100 m resolution) for 44 active volcanoes, integrating Sentinel-1 InSAR interferograms & coherence, SRTM DEM, and ERA-5 atmospheric humidity/temperature/pressure. Expert labels provide deformation type, intensity, temporal phase and free-text notes. Two tasks are defined, i.e., binary deformation detection and semantic segmentation, under single-epoch and three-frame settings. Baselines (ResNet-50, ConvNeXt, ViT, UNet, DeepLabv3, SegFormer) reveal the impact of atmospheric channels and short sequences. All data and code are released under MIT / CC-BY licences.

**Dataset Code Accessibility:**

Yes

**Dataset Code Comments:**

All data and code are released under MIT / CC-BY licences.

**Ethical Comments:**

Nil

**Ethical Considerations:**

No, there are no or only very minor ethics concerns

**Final Justification:**

I think authors addressed my concerns well.

**Limitations Weaknesses:**

1. Clarification on the Dataset's Importance
I have concerns regarding the significance of this dataset. Volcanic monitoring is typically most valuable when conducted in advance or in near real-time, as this enables timely forecasting and risk mitigation. In contrast, this dataset primarily focuses on historical events. The authors are encouraged to more clearly articulate the value of retrospective data in the context of model training, validation, or long-term volcanic pattern analysis, to justify its relevance and utility.

2. Inconsistencies in Spatial Resolution
The dataset aggregates data from multiple sources with varying spatial resolutions. Therefore, it may not be appropriate to refer to the dataset as uniformly having 100-meter resolution. The authors should explicitly address this limitation and clarify the methodology used to harmonize data across different resolutions, as this affects the dataset’s applicability and credibility.

3. Lack of Discussion on Model Transferability
The paper does not discuss the generalization ability of models trained using this dataset. Given that volcanoes can exhibit diverse morphological and temporal characteristics, it is important to assess whether models trained on one set of volcanoes can perform reliably when applied to others. A discussion or preliminary evaluation of this aspect would significantly strengthen the paper.

4. Citation Format Issues
Several references in the manuscript do not conform to the official NeurIPS citation format. Please revise the references accordingly to ensure consistency with the conference's style guide.

**Strengths Contributions:**

1. It is claimed that Hephaestus Minicubes is the first volcanic unrest monitoring dataset for machine learning.
2. Fully open data & code under MIT/CC-BY licences, enabling easy reproduction and extension.
3. Higher resolution & richer modalities: resolution improved from ≈333 m to ≈100 m; DEM and ERA-5 atmospheric layers added.

---

> ### Author Rebuttal · Authors · 2025-07-30
>
> We thank the reviewer for the insightful comments. We will try to address each point individually in the order they were presented.
>
> ## Clarification on the Dataset's Importance
> We agree that a better articulation of the proposed dataset motivation would be beneficial for the reader.
>
> Our work is focused on advancing early warning systems for volcanic activity. To do that we utilize ground deformation, a well-established precursory signal known to be statistically linked with eruptive events [1], which can be detected prior to the event [2]. Specifically, ground deformation can occur several weeks, and even years, preceding an eruption [1], **offering critical lead time for hazard assessment**, mobilising scientific teams to deploy sensing equipment on the ground and alert civil protection authorities. Consistent with established approaches in the field, as outlined in the Related Work section, we treat volcanic activity monitoring and early warning as a ground deformation detection problem.
>
> InSAR data from Sentinel-1, with a 6-day repeat-cycle, allows the systematic monitoring of ground deformation at a global scale without the need for in-situ instrumentation. These timescales are well-suited for tracking the onset and subsequent long-term evolution of volcanic unrest, supporting near-weekly monitoring of volcanoes and enhancing early warning, preparedness, and crisis response efforts.
>
> While our dataset is based on historical Sentinel-1 InSAR time series of past global volcanic unrest events, its **primary value lies** in supporting the **development**, training, and rigorous validation **of machine learning models for detecting volcanic unrest**. These models can then be applied to additional Sentinel-1 acquisitions to identify early signs of unrest at both known and previously unmonitored volcanoes.
>
> Moreover, retrospective analysis is essential for capturing the temporal evolution and variability of unrest signals, which often develop gradually over weeks to months prior to an eruption.
>
> The introduction of Hephaestus Minicubes is a significant step forward for the domain of volcanic unrest monitoring, as the curation of such large-scale, multi-modal, annotated datasets is highly laborious and expensive, while critical for the development and improvement of task-specific ML models.
>
> In summary, we believe this dataset is important for several reasons:
> 1. It captures volcanic unrest manifested as ground deformation—commonly a precursor to eruption—within timescales that align with known unrest-to-eruption intervals.
> 2. It supports early warning systems by providing critical information weeks to months in advance, enabling volcanologists and civil protection authorities to deploy ground-based instrumentation and implement disaster mitigation strategies.
> 3. It leverages Sentinel-1 InSAR, a rich, freely available, yet underutilized satellite data source, that allows the deployment of early warning applications.
> 4. Finally, we emphasize that such annotated datasets are rare and labor-intensive to produce, as they require expert interpretation of subtle geophysical signals. By making this dataset publicly available, we aim to accelerate research on automated monitoring systems that will ultimately contribute to earlier warnings and improved hazard mitigation.
>
> We will include this discussion into the introduction to better articulate the motivation behind the dataset and its relevance to the development of early warning systems.
>
> [1] Biggs, J., Ebmeier, S., Aspinall, W. et al. Global link between deformation and volcanic eruption quantified by satellite imagery. Nat Commun 5, 3471 (2014).
>
> [2] M. A. Furtney, M. E. Pritchard, J. Biggs, S. A. Carn, S. K. Ebmeier, J. A. Jay, B. T. McCormick Kilbride, and K. A. Reath, “Synthesizing multi-sensor, multi-satellite, multi-decadal datasets for global volcano monitoring,” Journal of Volcanology and Geothermal Research, vol. 365, pp. 38–56, 2018.
>
> ## Inconsistencies in Spatial Resolution
>
> InSAR phase and coherence products, provided at \~100-meter resolution, constitute the core input for volcanic unrest detection in our dataset. Additional variables, such as atmospheric data from ERA5, are acquired at coarser scale, and ensuring consistency across modalities, are resampled using bilinear interpolation to match the spatial resolution and grid of the InSAR products (\~100m × 100m). We will make this clear in Section 3 of the main text for the camera-ready version.
>
>
> ## Lack of Discussion on Model Transferability
>
> We adopted a temporal split in our benchmark to more closely reflect the conditions of real-world early warning systems. In operational settings, the volcanoes under observation and their locations are typically known in advance, allowing models to leverage historical events at those specific sites.
>
> That said, we fully agree that assessing how well models generalize to previously unseen volcanoes is an important consideration, both for understanding performance across diverse geological contexts and for enabling real world early warning systems to expand to new regions. To address this, we conducted a preliminary study to assess spatial generalization with the following experimental setup.
>
> We keep a temporal split for training and validation similar to the current benchmark, but hold out a subset of two **previously unseen volcanoes** specifically for the test set. These volcanoes were selected based on their activity frequency profiles, representative of frequent activity (0.55 positive/negative ratio) and infrequent activity (0.03). The final test set has an overall positive/negative ratio of 0.07, closely matching that of the original benchmark split (0.085), to ensure a fair basis for comparison. This experimental setup simulates a real-world scenario where the monitoring system is extended to new volcanoes, while effectively assessing the spatial generalizability of the existing benchmark.
>
> In this preliminary study, we evaluate a DeepLabV3 with a ResNet50 as backbone for segmentation tasks, and a ResNet50 for classification. Our initial results for single-timestep input are quite encouraging:
>
> - Classification: 87.5% F1 with atmospheric input /  86.98% F1 without atmospheric input
> - Segmentation: 75.22% F1 with atmospheric input / 74.55% F1 without atmospheric input
>
> While defining a representative spatial split is non-trivial due to the natural class imbalance across positive and negative instances as well as ground deformation types and event intensities, our initial results suggest that the models exhibit promising spatial generalization.
>
> We will extend this study, complementing it with the full range of experiments, along with an investigation on the impact of varying spatial splits (which affects the distribution of ground deformation types, event intensities, state of the volcano e.g rest, rebound, as well as positive/negative ratio). We will include this in the supplemental material and also discuss it in Section 5 of the main text.
>
> ## Citation Format Issues
> We will adapt citations to adhere to the NeurIPS guidelines in the camera-ready.

---

> > ### Comment · Reviewer_Tq9w · 2025-08-05
> >
> > I thank the rebuttal and efforts for the authors. I think authors have addressed my concerns already, and they agreed to revise the paper accordingly in the camera-ready version. Therefore, I think I will raise my score.

---

### Note · Authors · 2025-08-14

We thank all reviewers for their detailed and constructive feedback. The review process has been insightful and has led to a clearer, higher quality manuscript.

We are encouraged that reviewers recognized the key **strengths** of *Hephaestus Minicubes*: its contribution as the first large-scale,  multi-modal, ML-ready dataset for volcanic unrest monitoring; the richness of expert annotations; the comprehensive benchmark; the reproducibility enabled by fully open code and data; and the overall clarity of our work.

Reviewers also identified areas where additional analysis or clarification could further strengthen the basis of our work, points which we have **rigorously addressed**. Notably:
- **Generalization on Unseen Volcanoes**: We conducted experiments on the models’ generalization performance for unseen volcanoes, with promising preliminary results — *Classification*: 87.50% F1 (with atmospheric input) / 86.98% (without); *Segmentation*: 75.22% F1 (w) / 74.55% (w/o). These experiments will be further extended in the camera-ready version, with additional discussion on the impact of varying spatial splits (*Supplemental Material*).
- **Motivation and Clarity**: Building on our detailed responses of the reviewer–author discussions, we will adapt the script to include:
	-  a clearer articulation of the dataset’s motivation, emphasizing its role in early warning systems (*Section 1*)
	-  explicit description of resolution harmonization procedures (*Section 3*)
	-  expanded discussion of time-series results and limitations (*Sections 5 and 6*)
	-  clearer motivation behind the qualitative results and the role of atmospheric variables (*Section 5*)
	-  description of the resolution of annotator disagreement (*Supplemental Material*).
- **Formating and GPU Memory**: We will update citation formatting and enhance the reported GPU memory usage with specific batch size consumption (*Supplemental Material*).

Overall, we believe the review process has greatly contributed to improving the quality, depth and clarity of the original manuscript. We are confident that this submission will serve as a valuable asset to the NeurIPS conference, as it introduces a high quality dataset and benchmark for volcanic unrest monitoring, addressing a critical gap in current research. We are hopeful that *Hephaestus Minicubes* will support the development of robust algorithms for this domain, serving as an accessible, lasting resource for the community.

---

### Decision · Program_Chairs · 2025-09-18

**Decision:**

Reject

**Comment:**

All reviewers are favorable on this paper. After the rebuttal, the one negative reviewer increased his score to BA. The authors provided robust rebuttals to all reviews, but there was limited response from the reviewers perhaps because most of their scores were already positive. The proposed dataset tackles an important, interesting problem in early warning of volcanic eruptions where there are few AI-ready datasets.

In their rebuttals, the authors addressed the concern about generalization to unseen volcanos, and clarified a number of points which must be included in the final version.

===== FINAL UPDATE FROM DB Track PCs ====

The final decision for this paper has been taken by the program chairs after consultation with the SACs. All Senior Area Chairs have ranked papers according to the feedback from the AC during the review process. We decided to leave the original meta-review to reflect the opinion of the AC in light of the initial discussions with reviewers and SAC.